# The Role of the Nurse in Informed Consent to Treatments: An Observational-Descriptive Study in the Padua Hospital

Veronica Strini [1,*], Roberta Schiavolin [2] and Angela Prendin [3]

1  Clinical Research Unit, University-Hospital of Padua, via Giustiniani 2, 35128 Padua, Italy
2  Continuity of Care Service, University-Hospital of Padua, via Giustiniani 2, 35128 Padua, Italy; roberta.schiavolin@aopd.veneto.it
3  Independent Research, University-Hospital of Padua, via Giustiniani 2, 35128 Padua, Italy; aprendin@yahoo.it
*  Correspondence: veronicastrini@gmail.com; Tel.: +39-3393301622

**Abstract:** Background: The process to obtain valid informed consent in healthcare reflects many aspects. Healthcare professionals that take care of the patient must provide him all the necessary information and verify his understanding, considering individual characteristics. Nurses are one of the main participants in this process. Objective: This study assesses nurses' perceptions of their role in the informed consent process. Material and Methods: An observational study involving 300 nurses operating in 13 wards of the Padua Hospital, through the submitting of a questionnaire in the period November–December 2018. Results: The final sample is made up of 206 nurses—27 males (13.11%) and 179 females (86.89%). Work experience, on average 15 years, is significant in determining the answers to questions about opinions and experiences. Age is significant in determining how often nurses provide information to the patient's family members about the actions to be taken after discharge. The ward was decisive in the responses related to information provided to patients on the nursing care level and the actions to be taken after discharge, and the definition of the nurse's duties. Conclusions: The data collected show the need for interventions to reduce the causes of difficult that the nurse has in informing patients.

**Keywords:** informed consent; nurse role; nursing procedures; patient consent; hospital wards

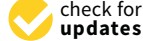



## 1. Background

Informed consent in healthcare is the process throughout patient decides freely and independently whether to start or continue the planned health treatment, after receiving specific information and made understandable to him by the doctor or health team based on their respective skills [1]. Information and informed consent play an essential role from a legal, as well as ethical point of view [2], as patients have the right to know their state of health, the diagnosis, and treatments available, the risks and probable benefits, and consequently choose any alternative [3]. Informed consent should be a process of mutually shared responsibility by the patient and the physician, ensuring adequate and relevant information that is well comprehended by the individual patient, and is used correctly for their decision-making [4]. It is very important to illustrate to the patient the possible risks in relation to complications and benefits, the presence of reasonable alternatives to the procedure, and the right of the patient to withdraw his consent at any time [5]. The accuracy of healthcare professionals in informing the patient should provide the necessary elements to make the most convenient and appropriate decision for his well-being [6]. Disclosure of information and informed consent are relatively new concepts in the patient–physician relationship. They are based primarily on the principle of autonomy, and they have many favorable practical advantages. However, the practical implementation of these requirements is fraught with difficulties, some of which can cause harm to the patient or be obstacles in fulfilling the moral obligation of beneficence [4]. There are numerous limitations in this process included: Patient understanding, potentially

alterations by factors, such as age, training, cognitive level, loss of control and level of anxiety, patient autonomy, how patients use information disclosed by healthcare professionals, and the time required to fully inform patients [7]. A central reference for the ethics of the health professions is represented by therapeutic self-determination, defined as the individual's right to decide on their own health, as a guarantee of their identity and dignity [8]. Information, in the past, has been considered as an exclusively medical aspect, but today must be considered as clinical, because it is common to many professions, integrated and focused on clinical-care pathways, rather than on individual services [9]. The practice of informed consent is multifaceted and presents complex legal and ethical challenges. Without a clear understanding of the underlying concepts and legal requirements that guide the process, informed consent may fall short of its goals. Nurses can be integral to the process while still practicing within their defined scope of practice. Nurses may serve to ensure patient comprehension, facilitate documentation of consent, address patient anxiety, and identify the appropriate surrogate decision-maker when needed [10]. Patient understanding and safety is the responsibility of all healthcare professionals [11].

The ideal completion of the informed consent process may be achieved if surgeons and knowledgeable nurses collaborate for the patient's good. With a broadened knowledge base on the realities and requirements of the informed consent process, nurses and surgeons may each contribute within their legal scope of practice to reduce the risk of litigation by fully meeting the legal obligations imposed by informed consent statutes [10].

For the nursing profession, the core-interest is the person and to respond effectively and competently to their nursing care needs, through a true involvement of individuals and families in the care processes that concern them [12]. Practices related to informed consent and the roles of nurses in the process are strongly influenced by the model of relationships between the different actors: Patients, their families, and healthcare professionals [13]. Nurses, who have intense interaction with patients, family members, and doctors, can ensure that patients' right to making an informed decision is respected. In daily practice, it is unclear whether nurses perform these roles in line with their professional service, or more doing the task of assisting doctors in providing patient care [13].

Although the literature argues that nurses should be advocates for this role, there is limited evidence that nurses are adequately prepared for this role, and that their role is adequately valued. To identify what nurses in preparation, need to be able to skillfully and confidently participate in the informed consent process in a highly hierarchical context, we explored the perception that nurses have regarding their involvement in different contexts of the same hospital, allowing comparative evaluations. In detail, nursing in Italy has reached important milestones in terms of training, decision-making autonomy, and responsibility, and in 2001 the university degree for nursing became the only path to access the profession. In recent years, a new generation of young university-trained nurses have been admitted into the world of work.

A primary reference for Italian nurses is the Code of Ethics for Nurses, and only a previous study has explored their practical adherence to the above standards [14] and the perception of the nursing role in this particular aspect.

*Objective*

To evaluate the perception of nurses regarding their role in informed consent to treatments.

## 2. Materials and Methods

This is a prospective descriptive observational study.

### 2.1. Sampling and Inclusion/Exclusion Criteria

The sample is made up of nursing staff operating in 13 wards of an Italian hospital: Cardiology, neurology, neurosurgery, hematology, general surgery, thoracic surgery, general medicine, Medical Clinic 1, Medical Clinic 5, geriatrics, urology, plastic surgery, and the specialized medical area ward. The final study sample is composed of 206 nurses.

## 2.2. Inclusion Criteria

- Active presence of nursing staff within the various teams during data collection;
- Voluntary participation in the study.

## 2.3. Exclusion Criteria

- Nursing staff of wards not included in the study: Intensive Care, Pediatrics, Operating Room, due to the lack of relevance with the proposed questionnaire.
- Lack of nursing staff within the team in the period dedicated to data collection;
- Not joining the study.

The questionnaire (Appendix A) was taken from the study by Ingravallo et al. [14]. The acquisition and permission for the use of the questionnaire were obtained through a direct request to the author. The questionnaire consists of an initial part of the personal data with information regarding gender, age, type of professional training, length of professional experience, activities in the ward or in an outpatient service. The questionnaire is divided into two units: Nurses' opinions and experiences. The questionnaires were administered by the researchers with dedicated and delayed weekly accesses from 15 November 2018 to 21 December 2018. Final access dedicated to the collection of the completed questionnaires was then carried out in each ward.

## 3. Statistical Analysis

Data collection, recording, and analysis were carried out using Microsoft Office Excel and R software. For the latter, "coin" and "exactRankTests" tool utilities were used.

Responses on a scale of intensity were dichotomized into two categories: "negative" (for nothing, little) and "positive" (enough, a lot). Frequency responses were dichotomized into the two categories "not frequently" (never, rarely, sometimes) and "frequently" (almost always, always). Continuous variables (i.e., age and length of professional experience) were dichotomized by the median values.

Descriptive statistics were presented as n (%) or mean (standard deviation (SD)), median, and range for categorical or continuous variables. A *p*-value < 0.05 was considered statistically significant.

## 4. Results

The final sample of respondents, from all wards (Table 1), was made up of 206 nurses (68.67%)—consisting of 27 males (23.11%) and 179 females (86.89%). The average age of the participants in years was 39.83 ± 9.78. The professional training of the sample (Table 2) was divided into three groups—64 nurses (31.07%) had obtained the Regional School Diploma, 135 nurses (65.53%) had a Bachelor's Degree, and 7 nurses (3.4%) had a Master's Degree. Two subgroups were added to these—17 participants (8.25%) had completed a Post Bachelor's Degree, while only one participant (0.49%) had completed a Post Master's Degree. The average working period corresponding to 15.71 ± 10.53, and the average working time in the ward is 10.48 ± 9.09.

### 4.1. Opinion Analysis

Between the five statements posed, statistical significance was found only in statement 7a, which appears to be statistically associated with the demographic and educational characteristics of the interviewees. In detail, it is observed that male subjects respond significantly more positively than female subjects. Furthermore, those who have been working for less time as nurses believe that the studies have been adequate to learn how to communicate information to the patient, compared to those who have been working longer. In particular, it is emphasized that there are no statistically significant differences between wards and the answers given to the opinion questions.

**Table 1.** Distribution of respondents by hospital ward.

| Ward | Frequency |
|------|-----------|
| Hematology | 21 (10.19%) |
| Neurosurgery | 20 (9.71%) |
| Neurology | 16 (7.77%) |
| Thoracic Surgery | 16 (7.77%) |
| General Surgery | 17 (8.25%) |
| Cardiology | 17 (8.25%) |
| Medical Clinic 1 | 13 (6.31%) |
| Urology | 11 (5.34%) |
| General Medicine | 13 (6.31%) |
| Specialized Medical Area | 10 (4.85%) |
| Medical Clinic 5 | 22 (10.68%) |
| Geriatrics | 10 (4.85%) |
| Plastic Surgery | 20 (9.71%) |

**Table 2.** Description of participant education.

| Total Number | 206 |
|--------------|-----|
| **Male/Female** | 27(13.11%) /179(86.89%) |
| **Professional Training** | |
| **Regional School Diploma** | 64 (31.07%) |
| **Bachelor's Degree** | 135 (65.53%) |
| **Master's Degree** | 7 (3.4%) |
| **Post Bachelor's Degree** | 17 (8.25%) |
| **Post Master's Degree** | 1 (0.49%) |
| **Middle Age ± Sd** | 39.83 ± 9.78 |
| **Working Years Average ± Sd** | 15.71 ± 10.53 |
| **Working Time in the Average ± Sd** | 10.48 ± 9.09 |
| **Work Activity in the Ward (on the Total Number)** | 206 (100%) |

*4.2. Experience Analysis*

There were significant differences between the answers given in the various wards (Table 3). In question 8a, which investigates how often the nurse has provided patients information about the nursing care plan, it was found that: The general surgery ward responded significantly with values lower than the scale compared to the hematology and neurosurgery wards. Compared to the neurosurgery ward, the neurology and urology wards also gave answers tending towards the minimum values of the scale.

**Table 3.** Different answers of the wards in relation to question 8a.

| Ward | Never | Rarely | Sometimes | Almost Always | Always |
|------|-------|--------|-----------|---------------|--------|
| Hematology | 4.76% | 19.05% | 23.81% | 23.81% | 28.58% |
| Neurosurgery | 5.00% | 10.00% | 25.00% | 25.00% | 35.00% |
| Neurology | 25.00% | 25.00% | 43.75% | 0.00% | 6.25% |
| Thoracic Surgery | 12.50% | 18.75% | 12.50% | 37.50% | 18.75% |
| General Surgery | 29.41% | 29.41% | 35.29% | 5.88% | 0.00% |
| Cardiology | 0.00% | 17.64% | 47.06% | 11.76% | 23.53% |
| Medical Clinic 1 | 23.08% | 23.08% | 23.08% | 15.38% | 15.38% |
| Urology | 27.27% | 45.45% | 18.18% | 9.09% | 0.00% |
| General Medicine | 15.38% | 30.77% | 30.78% | 23.08% | 0.00% |
| Specialized Medical Area | 10.00% | 40.00% | 30.00% | 0.00% | 20.00% |
| Medical Clinic 5 | 22.73% | 22.73% | 31.82% | 13.64% | 9.09% |
| Geriatrics | 20.00% | 20.00% | 20.00% | 30.00% | 10.00% |
| Plastic Surgery | 10.00% | 10.00% | 35.00% | 45.00% | 0.00% |

The responses to question 8d (Table 4), which investigates how often nurses provide patients information about the disease they were suffering from, are relevant. In the

Medical Clinic 5 ward, nurses responded with significantly more negative scale values than in the cardiology and hematology ward. In the hematology ward, they responded significantly with higher values of the scale than in the neurology, thoracic surgery, general surgery, Medical Clinic 1, and Medical Clinic 5 wards.

**Table 4.** Most significant response differences in relation to question 8d.

| Ward | Never | Rarely | Sometimes | Almost Always | Always |
|---|---|---|---|---|---|
| Hematology | 4.76% | 23.81% | 33.33% | 14.29% | 23.81% |
| Medical Clinic 5 | 40.90% | 31.28% | 2.73% | 4.55% | 0.00% |

Question 8g (Table 5) investigates how often the nurse provided information to the patient about what to do at discharge: The nurses of the plastic surgery ward responded with significantly higher values than the wards of Medical Clinic 5, general surgery, and neurology.

**Table 5.** Significant differences in response to question 8g.

| Ward | Never | Rarely | Sometimes | Almost Always | Always |
|---|---|---|---|---|---|
| Hematology | 0.00% | 9.52% | 19.05% | 38.10% | 33.33% |
| Plastic surgery | 0.00% | 0.00% | 20.00% | 55.00% | 25.00% |
| Medical Clinic 5 | 9.09% | 22.73% | 45.45% | 18.18% | 4.54% |
| General surgery | 5.88% | 41.18% | 23.53% | 29.51% | 0.00% |
| Neurology | 18.75% | 18.75% | 56.25% | 6.25% | 0.00% |

As regards the age of respondents, it was significant that in question 8c how often nurses provide information about an invasive nursing maneuver before carrying it out, with increasing age nurses respond with statistically more negative values than the scale.

There are significant differences between wards. Question 9b asks how much it facilitated the work of the nurse to provide information to the patient about the nursing care plan. The nurses in the general surgery ward responded more negatively than the scale compared to their colleagues in hematology and neurosurgery.

### 4.3. Analysis of the Difficulty in Communicating Information to Patients

The analyses did not show statistically significant associations between the difficulties expressed by the nurse in providing information to the patient requested in question 10 and the demographic and educational characteristics of the interviewees. In question 11, related to the causes of difficulties in providing information to patients, it was found that: Subjects respond differently based on education, gender, and ward. The following considerations emerged from the analyses:

-   Those who do not have a Master's Degree more easily select option 11, which focuses on the lack of personal training regarding how to communicate with patients.
-   Those with a Master's Degree are more likely to select answer 11f, which indicates the lack of sufficient information about the patient's diagnostic-therapeutic project.
-   Men more than women, those with more work experience, and those with a Master's Degree are more likely to select option 11a, which indicates the patient's poor ability to understand information.

This option was specially selected for the geriatrics ward (Table 6).

**Table 6.** Response rate to question 11, with option 11e in the various wards.

| Ward | % Not Selection | % Selection |
|---|---|---|
| Hematology | 80.96 | 9.04 |
| Neurosurgery | 95.00 | 5.00 |
| Neurology | 87.50 | 12.50 |
| Thoracic surgery | 81.25 | 18.75 |
| General surgery | 76.47 | 23.53 |
| Cardiology | 82.35 | 17.65 |
| Medical Clinic 1 | 92.31 | 7.69 |
| Urology | 72.72 | 27.28 |
| General medicine | 84.62 | 15.58 |
| Specialized Medical Area | 90.00 | 10.00 |
| Medical Clinic 5 | 77.27 | 26.73 |
| Geriatrics | 30.00 | 70.00 |
| Plastic surgery | 85.00 | 15.00 |

*4.4. Written Consent*

Question 13b asked how often the nurse asked the patient to sign the written consent for the execution of surgery or instrumental investigation. It appears that in the hematology ward, the answer was higher than in neurosurgery, thoracic surgery, general surgery, and Medical Clinic 5.

*4.5. Information to Family Members*

Question 14c asked, "How often did you provide the patient's family with information about: What to do after discharge?". Regarding work experience, it should be noted that as the years of work increase, the question is answered with significantly more positive values. Particularly in the hematology ward, the nurses responded with significantly higher values on the scale than their colleagues in the general surgery, cardiology, general medicine, and Medical Clinic Wards 5. In the Medical Clinic Ward 5, the responses had significantly lower values compared to the wards of urology, plastic surgery, and geriatrics. In the plastic surgery ward, they responded with significantly higher values than their colleagues in the cardiology and general medicine. Finally, there appears to be no significant associations for question 15—"How often did you provide family members with information that you did not provide to the patient?".

*4.6. Role of the Nurse within the Team*

There were associations fund regarding the education level of the nurse and ward they worked on. In particular, the response in the neurology ward was significantly lower than in the neurosurgery ward. Nurses who had obtained a Master's Degree gave answers' rate lower compared to the values of those who have a Bachelor's Degree.

*4.7. Analysis of Marginal Responses*

Questions 8a asked, "How often did you provide patients with information about the nursing care plan?", 14b asked, "How often did you provide the patient's family with information about: The nursing care plan?" and 14c, mentioned above, no significant deviation from the central value of the scale was observed. In the case of question 7a—"The course of study adequately prepared me to communicate information to the patient."—no statistically significant differences were found.

**5. Discussion**

This study investigated the perception that nurses have regarding their training, provided by their curriculum, in communicating information to patients. As a first result, it emerged that male subjects (13.11% of nurses) responded significantly more positive values (*p*-value = 0.002) than female subjects (86.89%). As a second result, it emerged that nurses with less work experience believed they have adequate preparation to inform

patients: Their responses found significantly more positive values (*p*-value = 0.000). It can be deduced that nurses with less work experience consider the training they have received adequate to communicate information to the patient compared to those who have worked longer. The data did not show statistically significant associations with the level of education and the young age of the nurses. On the contrary, in the study carried out by Ingravallo et al., in addition to the reduced work experience (*p*-value = 0.042), associations with young age (*p*-value = 0.016) and university education (*p*-value = 0.002) were found [15]. Nurses with these characteristics are more confident that their training is adequate to provide information to patients. In general, the answers to this question do not differ from significantly positive or negative values (*p*-value> 0.05), suggesting the need for updating within universities and hospital wards, involving more nurses who have been operating for a longer time [10,11]. The years of work within the various ward (10.48 $\pm$ 9.09), the achievement of Post Bachelor's Degree (8.25%) and Post Master's Degree (0.49%) were not significant in determining the answers to questions of opinion and experiences. As reported in 2010, both graduate and non-graduate nurses may have difficulty in their own ethical responsiveness to address areas of interest in all healthcare contexts [12]. In his investigation in four different surveyed regions of the United States, Ulrich found that informed consent remains one of the most frequently cited ethical issues occurring in daily clinical practice [16]. As reported by Axson in 2017, this remains the same today: Thirty percent of our participants disagree with the statement that their undergraduate education has prepared them to actively engage in informed consent processes. Despite this lack of preparation in university education, three-quarters of nurses in the study reported that when they are unsure of one component of the informed consent process, they know which resources they are available on their unit to assist them [11]. In Ingravallo's study of 2017 [14], regarding the item on whether nurses had any difficulty in providing information to patients, both Korean and Italian nurses reported only slight difficulties, correlated to lack of time or opportunity during work hours, patients' inability to understand the information and inadequate information regarding the patient's treatment plan. The same author says that there is inadequate focus on communication skills in both the medical and nursing educational curriculums.

The present study has shown, with statistically significant values (*p*-value < 0.05), that nurses in their practice inform and require the patient's consent before carrying out invasive nursing maneuvers. This data agrees with what is expressed by Ingravallo et al. in which 94% of nurses claim to inform patients about these procedures [15]. However, according to the nurses, providing this information did not facilitate them in carrying out their work: When they were asked to indicate whether informing the patient about the invasive nursing procedures implemented facilitated the nursing practice, they responded with values significantly lower than the scale items (*p*-value < 0.05). Furthermore, a statistically significant association (*p*-value < 0.05) emerges between the increase in the age of nurses and the frequency with which they inform patients before performing invasive nursing maneuvers: With increasing age, the nurses provide information, with respect to the invasive nursing maneuvers they implement, with less frequency to patients. This may be because with increasing age and professional experience, nurses are less motivated to inform patients about the services they are about to perform. This is because they often perform their duties in a routine way, maintain their knowledge is rooted, and show little interest in updating [17]. It can be deduced that they do not provide information on this, since they believe that patients cannot understand them. In fact, from the analyses carried out, this aspect is indicated as one of the main problems in providing information to patients (option selected by 39 nurses); this data emerged predominantly (*p*-value < 0.05), especially for those nurses who have more work experience. In general, it emerges that nurses claim to frequently inform patients about what to do after discharge, answering the question with significantly higher values than the items on the scale (*p*-value < 0.05). This finding agrees with the literature in the study by Ingravallo et al.—69% of nurses provide this information to patients [15]. In particular, patients were more informed about

the actions they must carry out after discharge within in the wards of hematology (almost always: 38.10%, always: 33.33%) and plastic surgery (almost always: 55%, always: 25%). On the contrary, in Medical Clinic 5 (never: 9.09%, rarely: 22.73%), neurology (never: 18.75%, rarely: 18.75%) and in general surgery (never: 5, 88%, rarely: 41.18%) patients are not adequately informed.

In the Aveyard focus group study [18], many nurses expressed the view that consent was implied prior to nursing care procedures if the patient has not objected. It was recognized that much of the routine nursing care would fall under this concept of implicit consent if a minimal explanation is provided to the patient. However, according to Cole [19], the use of implied consent still relies on the nurse to render some of the decisions for the patient rather than for the patient who has the autonomy to make their own decisions.

The previous theme is often connected with the practice of providing information by nurses to patients' relatives. Colleagues in hematology respond with significantly more positive values than colleagues in general surgery, cardiology, general medicine, and Medical Clinic 5. Furthermore, the frequency with which nurses inform family members about this is strongly correlated ($p$-value < 0.05) to their work experience: As they increase their working years, they claim to provide this information more frequently than their colleagues with less work experience. It can be assumed that more experienced nurses are more aware of the key role that family members play in taking care of the patient once discharged, so they take the time to inform them adequately.

Ingravallo affirmed, in 2017, that partial or no disclosure of diagnosis and prognosis and family-centered decision-making remain common, especially among older patients and patients with cancer [14]. This is mainly attributed, in Italy, to a culture that prioritizes family and community ties over individual 'self-governance'.

Regarding the definition of the nurse's tasks in informing the patient within the team, in this study the nurses responded with significantly lower values than the items on the scale ($p$-value < 0.05). This data disagrees with what emerged in the study by Ingravallo et al., in which 59% of nurses affirmed the presence of an adequate definition of tasks within the teams they considered [15]. This difference in response can also be found in the causes of difficulties in providing information to patients (especially in the geriatrics ward: 70%). Nurses with Master's Degrees most likely attribute it to the lack of sufficient information about the diagnostic project of the patient and the patient's poor ability to understand information, the latter being selected more by male participants.

In addition to personal characteristics, cultural background, and educational background, the relationship between nurse and patient is influenced by many other factors that are beyond the nurse's control [20]. For example, the care environment [21] and organizational factors, such as a high workload: These may be poorly perceived by patients, but are highly perceived by nurses, creating discrepancies between patient and nurse expectations in behaviors care [22].

In the present study, nurses were given the opportunity to express personal considerations on the matter. The most-reported difficulties are related to: The lack of an appropriate place to inform patients and caregivers that guarantees privacy; the lack of time to dedicate oneself to informing the patient correctly (reported by 64 nurses); inform the nursing staff of the diagnostic and therapeutic process of the patients (reported by 54 nurses). It is important that there is little communication between nurse and doctor for organizational reasons: Many times, the nurse is forced to provide information of medical relevance to the patient and family members. The present study emphasizes that family members ask nurses for information that is not within their competence with statistically significant values ($p$-value < 0.05). Studies show that patients believe nurses are well informed about their medical conditions, medications, and treatments [10,22]. Nurses, in fact, consider themselves more accessible in providing information to patients than doctors [23]: This is due to the limited time that doctors must talk to patients and family members [24,25]. Moreover, Italian nurses are currently still facing fundamental changes in their roles and responsibilities, and this may result in uncertainty about their role and their level of auton-

omy within a team [15]. Furthermore, in this study appears the need to better define the role of nurses in information delivery to patients in Italy.

## 6. Conclusions

In daily practice, nurses take a fundamental role during the informed consent process. As patient advocates and direct care providers, nurses have a unique opportunity to meaningfully advocate for mutual decision-making—a process that promotes patient autonomy, comprehension, and self-determination. Empowered by a comprehensive understanding of the informed consent process, nurses can serve in that advocacy role without running the risk of practicing outside their professional scope by assuming the responsibility for consenting the patient. The ideal completion of the informed consent process may be achieved if surgeons and knowledgeable nurses collaborate for the patient's good. With a broadened knowledge base on the realities and requirements of the informed consent process, nurses and surgeons may each contribute within their legal scope of practice to reduce the risk of litigation by fully meeting the legal obligations imposed by informed consent statutes [10].

The present study could be extended to several realities to have a larger sample, compare the results obtained and make them generalizable to Italian nursing practice. Furthermore, it would be useful to carry out studies, parallel to this, which also involve doctors and patients from different wards, to compare the results and have a global concept of informed consent in the hospital environment. This provides useful data to raise awareness among Healthcare Professionals and Institutions regarding the difficulties still present.

## 7. Limits

The following limitations were found in this study:

- The presence of low sample size by ward and educational qualification;
- The results of the study cannot be generalized to all Italian hospital wards, nor to care settings outside the hospital.

## 8. Ethical Aspects

Before starting the data collection, authorization was requested from the wards involved in the study. The purposes of the study and the methods of data collection were explained to the participants. The privacy and anonymity of the participants were guaranteed, respecting the voluntary nature of participation in the study.

**Author Contributions:** Conceptualization: V.S., R.S. and A.P.; Methodology, V.S., R.S. and A.P.; Software, V.S., R.S. and A.P.; Validation, V.S., R.S. and A.P.; Formal Analysis, V.S., R.S. and A.P.; Investigation, V.S., R.S. and A.P.; Resources, V.S., R.S. and A.P.; Data Curation, V.S., R.S. and A.P.; Writing—Original Draft Preparation, V.S., R.S. and A.P.; Writing—Review and Editing, V.S., R.S. and A.P.; Visualization, V.S., R.S. and A.P.; Supervision, V.S., R.S. and A.P.; Project Administration, V.S., R.S. and A.P.; Funding Acquisition, V.S., R.S. and A.P. All authors contributed to the same and review of the work. All authors have read and agreed to the published version of the manuscript.

**Funding:** This research received no external funding.

**Institutional Review Board Statement:** Ethical review and approval were waived for this study, due to the fact that the participants were not patients, but healthcare professionals, and that anonymity and privacy were guaranteed at every step of the study.

**Informed Consent Statement:** Informed consent was obtained from all subjects involved in the study.

**Data Availability Statement:** The authors confirm that the data supporting the findings of this study are available within the article. Raw data or tables omitted in this article that support the findings of this study are available from the corresponding author, upon reasonable request.

**Acknowledgments:** We thank Giosuè Bacchin for his precious assistance for document translation.

**Conflicts of Interest:** The authors declare no conflict of interest.

**Appendix A. Patient Information and Consent Questionnaire**

*Appendix A.1. Patient Information and Consent*

Questionnaire on the views and experiences of nurses in the hospital

1. Age _______
2. Sex M☐  F☐
3. Professional training
☐ Regional School Diploma  ☐ Bachelor's Degree ☐ Master's Degree
☐ Post Bachelor's Degree  ☐ Post Master's Degree ☐ PhD
☐ Other (specify) _____________________
4. How long have you been working as a nurse? _________ years (if less than 1 year specify the months)
5. How long have you been working in this ward? __________ years (if less than 1 year specify the months)
6. His work activity takes place mainly in: Ward ☐ Outpatient/DH ☐

Appendix A.1.1. Part 1: Opinions

7. How much do you agree with each of the following statements?

| | | | | |
|---|---|---|---|---|
| **a. The course of study has adequately prepared me to communicate information to the patient** | For nothing ① | Little ② | Enough ③ | A lot ④ |
| **b. The nurse can provide the patient with information, even if it is not strictly his competence, if he has the knowledge to do so** | For nothing ① | Little ② | Enough ③ | A lot ④ |
| **c. The nurse should provide information to the patient only if he requests it** | For nothing ① | Little ② | Enough ③ | A lot ④ |
| **d. If the patient does not express his disagreement with an invasive nursing procedure (e.g., insertion of a venous catheter), the nurse can proceed by assuming consent** | For nothing ① | Little ② | Enough ③ | A lot ④ |
| **e. Providing the patient with complete information on nursing care can expose the nurse to a greater risk of complaints** | For nothing ① | Little ② | Enough ③ | A lot ④ |

Appendix A.1.2. Part 2: Experiences

We ask you to answer the next questions considering what your relationship has been in the last month with patients who were conscious and at least partially able of receiving information and expressing their consent

8. How often did you provide information to patients about:

| | **Never** | **Rarely** | **Sometimes** | **Almost Always** | **Always** |
|---|---|---|---|---|---|
| a. The nursing care plan | ① | ② | ③ | ④ | ⑤ |
| b. The therapies he/she gave him | ① | ② | ③ | ④ | ⑤ |
| c. An invasive nursing maneuver (e.g., insertion of a venous catheter), before performing it | ① | ② | ③ | ④ | ⑤ |
| d. The disease they were suffering from | ① | ② | ③ | ④ | ⑤ |
| e. A diagnostic test they had to do | ① | ② | ③ | ④ | ⑤ |
| f. A surgery they had to undergo | ① | ② | ③ | ④ | ⑤ |
| g. What to do after discharge | ① | ② | ③ | ④ | ⑤ |

9. How much did it facilitate your work to provide information regarding:

| | For Nothing | Little | Enough | A Lot |
|---|:---:|:---:|:---:|:---:|
| a. Therapy | ① | ② | ③ | ④ |
| b. The nursing care plan | ① | ② | ③ | ④ |
| c. An invasive nursing maneuver (e.g., insertion of a venous catheter), before performing it | ① | ② | ③ | ④ |

10. Was it difficult to provide information to patients?

| For nothing ① | Little ② | Enough ③ | A Lot ④ |
|---|---|---|---|

11. If you answered a lot or enough, can you indicate what were the main causes of difficulty? (can give more than one answer)

[a] Little personal training on how to communicate with patients

[b] Lack of opportunities (hospitalization too short, patient managed only for a short time)

[c] There is little time available during the shift

[d] Too much complexity of the information to be provided

[e] Poor patient's ability to understand information

[f] Not having sufficient information about the patient's diagnostic-therapeutic project

[g] Other ______________________

12. How often did patients or their family members ask you for information that was not a nursing concern?

| Never ① | Rarely ② | Sometimes ③ | Almost Always ④ | Always ⑤ |
|---|---|---|---|---|

13. How often:

| | Never | Rarely | Sometimes | Almost Always | Always |
|---|:---:|:---:|:---:|:---:|:---:|
| a. You asked the patient for consent before performing an invasive nursing maneuver (e.g., insertion of a venous catheter) | ① | ② | ③ | ④ | ⑤ |
| b. You asked the patient to sign the written consent to perform surgery or an instrumental investigation | ① | ② | ③ | ④ | ⑤ |

14. How often did you provide the patient's family with information about:

| | Never | Rarely | Sometimes | Almost Always | Always |
|---|:---:|:---:|:---:|:---:|:---:|
| a. The pathology | ① | ② | ③ | ④ | ⑤ |
| b. The nursing care plan | ① | ② | ③ | ④ | ⑤ |
| c. What to do after discharge | ① | ② | ③ | ④ | ⑤ |

15. How often did you provide family members, separately, with information that you did not provide to the patient?

| Never ① | Rarely ② | Sometimes ③ | Almost Always ④ | Always ⑤ |
|---|---|---|---|---|

16. Was the nurse's duties in providing information to the patient well defined in the medical-nursing team?

| Never ① | Rarely ② | Sometimes ③ | Almost Always ④ |
|---|---|---|---|

17. Is there something else that you want to point out on the issues of information and consent or something you want to specify about the answers you gave to the questionnaire?

_______________________________________________________________________________

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
