# Peer review of "The Role of the Nurse in Informed Consent to Treatments: An Observational-Descriptive Study in the Padua Hospital"

_clinpract, doi:10.3390/clinpract11030063_

Round 1
Reviewer 1 Report
I congratulate the Authors the main idea of this study. I really enyoy the studies about the informed consent. The introduction is written fine, but than I cannot find any positive remarks. In my opinion, it is only the opinion this study does not bring anthing to the existing knowledge.
You could improve this article by making more comparisons, the reasons that would highlight the results. Now it looks very poor in trems of methods and the what is the purpose to perform ythis study?
Author Response
We added an opening paragraph specifying the reason for conducting the study (line 72-88).
We have added more comparisons to the discussion than the existing literature (line 312-328; 359-365; 375-378; 407-410)
Reviewer 2 Report
Thanks for your work. However, there seems to be insufficient discussion based on the results from this study. So we can't really grasp the implications of this study for us.
Author Response
We have added more comparisons to the discussion than the existing literature (line 312-328; 359-365; 375-378; 407-410)
Reviewer 3 Report
This study is the role of the nurse in informed consent to treatments. It is a good topic for clinical nurses. However, it is unclear the theoretical background of the topic and the necessity for study. Currently, in many countries, nurse have the role of informed consent to treatments as increasing the number of nurse practitioners and specialists. Why this is a problem and which part is a problem according to the medical environment of each country. You should be described in detail about it. In particular, statistical analysis is necessary to explain and understand the research problem to the reader. However, this paper also has many insufficiency in the statistical part. The format of the table presented to readers is also very poor.
Author Response
We added an initial paragraph on reason for conducting the study (line 72-88).
We increased informations on paragraph concerning statistical analysis in line with the presentation of the results (line 178-185).
We reviewed the format of the graphics, we will then ask the publisher if necessary changes and whether to leave the qustionary along the text or insert it as a separate attachment.
Reviewer 4 Report
The issue of the role of the nurse in informed consent to treatment is not very often to find in research works. Mostly we can read ethical problems in nursing from the general point of view. Therefor this article is very interesting and contributing especially for the practice with its concrete situations where the nurse should obey ethical principles mainly communication skills, which are time consuming but necessary.
I would recommend mentioning the number of questions in the questionnaire. Although the questionnaire is attached, it is not clear, whether it is whole or just a part of it. I don´t see the reason to have it in the article.
Some of the tables are redundant (1,4,6) because they don´t underpin the results with any special way.
The title of the Table 2 should be Description of participant education
the discussion is based on Ingravallo work. Is there any other research on this issue to be discuss with in the discussion?? Some parts of the discussion is more like results description with no reaction on them – what does the finding means for the practice.
Author Response
I would recommend mentioning the number of questions in the questionnaire. Although the questionnaire is attached, it is not clear, whether it is whole or just a part of it. I don´t see the reason to have it in the article.
We will ask the editor if it can be attached separately or leave it in the text or remove entirely having made reference to the original article.
Some of the tables are redundant (1,4,6) because they don´t underpin the results with any special way.
We have considered inserting these tables to underline certain results that are present verbatim. If deemed necessary, we remove the 4 and 6, while I consider to keep the 1 for completeness of the information with respect to the textual part only and to have an initial graphic reference of all the departments considered in the study.
The title of the Table 2 should be Description of participant education
Right. We have changed it.
the discussion is based on Ingravallo work. Is there any other research on this issue to be discuss with in the discussion?? Some parts of the discussion is more like results description with no reaction on them – what does the finding means for the practice.
We have added more comparisons to the discussion than the existing literature (line 312-328; 359-365; 375-378; 407-410)
Round 2
Reviewer 1 Report
accept in a persent form
Reviewer 2 Report
Thank you for your contribution.
The description of thIs paper has been improved more systematically.
Despite the limitations of the study, it seems to have been written to more convincingly empathize with the claims of this paper.
Reviewer 3 Report
Thank you for your revision file.